# Injury epidemiology and emergency department length of stay in trauma hospital in Addis Ababa, Ethiopia

**Hailemichael Mulugeta**[1,2]*, **Ayalew Zewdie**[3], **Tesfaye Getachew**[3], **Wakgari Deressa**[1]

1 Department of Preventive Medicine, School of Public Health, College of Health Sciences, Addis Ababa University, Addis Ababa, Ethiopia, 2 Department of Public Health, College of Health Sciences, Debre Berhan University, Debre Berhan, Ethiopia, 3 Department of Emergency and Critical Care, AaBET Hospital, St. Paul's Hospital Millennium Medical College, Addis Ababa, Ethiopia

* hailumary464@gmail.com

**Data Availability Statement:** All relevant data are within the paper.

**Funding:** The author(s) received no specific funding for this work.

## Abstract

### Background

Injuries are a major cause of health problems in low- and middle-income countries than in high-income nations. This study aimed to describe injuries and identify factors associated with the emergency department (ED) length of stay (LoS).

### Methods

This study was conducted at the Addis Ababa Burn, Emergency, and Trauma (AaBET) hospital in Addis Ababa, Ethiopia. All injured patients who visited the ED between April 1, 2021, and March 30, 2022, were included in the study. Data were collected through a retrospective medical record review. Descriptive statistics were used to present the data and a multivariable binary logistic regression model was used to assess factors associated with LoS in the ED.

### Results

A total of 6991 (86.8%) injured patients were studied out of 8055 patients who visited the ED. The majority of the patients were male, 5184 (74.2%), and aged between 18–44 years, 4368 (62.4%). The most common mechanism of injury was road traffic accidents, 2693 (38.5%), followed by falls, 1523 (21.8%). The most commonly reported injured body parts were the head (2179, 31.2%). Of the total injured patients, 101 (1.8%) died. The LoS in the ED was greater than 24 hours for 24.4% (95% CI: 23.3–25.5) of the injured patients, with a mean duration of 2.51 (SD = 5.18) days. Factors significantly associated with a prolonged LoS (>24 hours) in the ED included being residing in outside Addis Ababa in Oromia [AOR: 1.61, 95% CI (1.38–1.88)], Amhara [AOR: 1.56, 95% CI (1.14–2.14)] and other [AOR: 3.93, 95% CI (2.73–5.64)], male [AOR: 1.30, 95% CI (1.09–1.54)], aged 60 years or older [AOR: 1.38, 95% CI (1.03–1.85)]; sustaining injuries from road traffic accident [AOR: 2.43, 95% CI (1.19–4.94)], being triaged to orange [AOR: 3.03, 95% CI (2.40–3.83)] and red zones [AOR:3.37, 95% CI (2.65–4.28)]; having fracture injuries [AOR: 1.95, (1.34–2.83)]; and experiencing contusions and crushing injuries [AOR: 2.63 (1.57–4.42)].

**Competing interests:** The authors have declared that no competing interests exist.

## Conclusions

Injuries are the major share of cases among ED patients at AaBET hospital. Patients were staying in the ED for longer than the recommended 24 hours by the Ethiopian health system. Intervention measures focusing on injury prevention are necessary to reduce the patient burden on the hospital and strengthen the hospital's capacity to reduce prolonged LoS.

## Introduction

Injury is a major cause of morbidity, disability and mortality [1]. In 2019, more than four million people died due to injuries, accounting for 8% of the global total deaths [2]. Approximately, 90% of these fatalities occur in low- and middle-income countries (LMICs). Despite significant declines in injury fatality rates in high-income countries (HICs), many LMICs have seen an increase in injury-related deaths, mainly due to road traffic injuries (RTIs), interpersonal violence, war, and self-inflicted injuries [2]. In Ethiopia, injury is the major cause of morbidity, disability, and mortality [3, 4]. It accounts for about 8% of all-cause mortality in Ethiopia, which is more than the combined number of deaths from tuberculosis and malaria [4]. According to a recent study on mortality at six health and demographic surveillance system (HDSS) sites in Ethiopia using the verbal autopsy technique, injuries accounted for about 6% of all-cause deaths. The majority of deaths from injuries occurred at home (47%) and at the site of the incident (38%), and 15% occurred in the health facilities [5]. More than 75% of injury deaths in Ethiopia are caused by RTIs, falls, drowning, self-harm, and interpersonal violence [4, 5].

Lessons learned over recent decades have shown that injuries are preventable public health problems through cost-effective preventive strategies [6]. In HICs, successful preventive measures and effective trauma care have resulted in significant reductions in injury-related deaths and disability [7]. However, injuries have remained a considerable cause of morbidity and mortality in LMICs as they received less attention from governments and donors in these countries compared to other public health problems [8]. Many countries have developed trauma registries to support and monitor the implementation of injury prevention and trauma care [9]. A trauma registry is a hospital-based standardized information system that characterizes injured patients with injury history, pre-hospital care, demographic, clinical, disposition, and outcomes on an ongoing basis [10]. The data can be used to monitor the quality of trauma care, allocate resources, evaluate the impact of performance improvement on the quality of care, and improve injury prevention [9].

However, most LMICs lack injury surveillance systems that can aid in the development of effective prevention and trauma care programs [6]. Recently, many countries in Sub-Saharan Africa (SSA) have attempted to establish hospital-based trauma registries to address these problems [11–13]. These countries have established trauma registries to collect data on injury events, trauma care, and patient outcomes. The establishment of these trauma registries is a timely step towards reducing the burden of injuries. This is done despite resource constraints such as a lack of trauma registry, trained healthcare workers, insufficient funding, and other competing priorities [6]. Several obstacles remain in the way of realizing the full potential of trauma registries to inform injury prevention, mitigate risks, and improve the standard of trauma care in SSA.

Despite the high burden of injury in Ethiopia, data on patterns and outcomes of injuries are limited. There are no trauma registries for collecting and analyzing injury data, which limits

the understanding and improvement of trauma care in the country. Only a few hospitals in the country have prospectively assessed the burden of injuries by establishing trauma registries for research purposes in public hospitals [13, 14]. Most hospitals use a mix of paper-based and electronic medical records as part of the national Health Management Information System (HMIS). Despite its limitations, retrospective study of hospital medical records is inexpensive and has provided useful information about the epidemiology and outcome of injuries [13, 15]. However, such data have been under-utilized in Ethiopia, with just a few studies collecting and analyzing injury data from the hospital's routine HMIS [16–20]. Hence, there are considerable gaps in understanding burden and characteristics of injuries in the country. This study aimed to describe injuries and identify factors associated with ED LoS of patients who presented to AaBET hospital in Addis Ababa. Such data will provide information on the burden of injury at the hospital and will serve as a baseline and reference for establishing and testing trauma registries in hospitals across the country.

## Methods

### Study setting

This study was conducted at Addis Ababa Burn, Emergency, and Trauma (AaBET) Hospital, a public hospital affiliated with St. Paul's Hospital Millennium Medical College in Addis Ababa City. The city has an estimated population of four million and is home to about 17% of Ethiopia's urban dwellers, but it only accounts for 3.3% of the country's projected 120 million people in 2022 [21]. The administrative division of Ethiopia consists of nine regional states and two chartered cities (Addis Ababa and Dire Dawa). The regional states include Amhara, Oromia, Southern Nations, Nationalities, and Peoples' Region (SNNPR), Tigray, Afar, Somali, Harari, Gambella, and Benishangul-Gumuz. The Oromia region, which is the largest in the country, surrounds Addis Ababa. Recently, two Sidama and Southwest Ethiopia regional states separated from the SNNPR and formed their own new states. Since the residential addresses of the majority of the patients were recorded in the patients' logbooks using the former SNNPR, we have used this regional name for this study.

AaBET Hospital is the main emergency and trauma referral hospital in Ethiopia, providing specialized and sub-specialized trauma care to both patients presenting primarily and those referred from other hospitals and health facilities in Addis Ababa, the surrounding Oromia areas and other parts throughout the country. The hospital has 250 inpatient beds and 14 Intensive Care Unit beds, serving as Ethiopia's main trauma care center. Very few studies have used the hospital's trauma surveillance system to assess the burden of injury and characterize trauma patients that present to the hospital [22–26].

AaBET Hospital provides 24-hour emergency and critical care services, orthopedic services, neurosurgery services, general surgery services, and burn care [26]. The ED has four zones (or wards) to treat patients according to the Emergency Severity Index scale [22]. These zones are identified by color-coded triage zones, which include green, yellow, orange, and red. Patients arriving at the ED are first assessed by emergency triage officers (physicians/nurses). Those patients, who are stable or have mild/minor severity are triaged to the yellow/green zone, while those with moderate severity are triaged to the orange zone. The red ward is used for severe patients who require extra care and stabilization.

There were 12 physicians, 8 interns, and 24 nurses during the day; and 8 physicians, 3 interns, and 24 nurses at night during the study period in the ED. Every month, more than 500 trauma patients are seen at the ED of the hospital. Patients are initially assessed and treated in the ED zone before being discharged, transferred to another department for admission, or referred to another hospital. The ED provides 24-hour emergency and critical care services.

Routine admission and management procedures are carried out upon the arrival of the patient in the ED [26].

Data recording was started as soon as the patient received a medical chart from the ED. All records for every patient are kept in a folder that also includes several charts that are crucial for documenting each activity during the course of care and treatment. The chart follows a patient's journey from one zone or department to another within the hospital. Then, the nurses at the ED record all of the data from the chart into the patient logbook as part of the HMIS. Then, data from the logbook are manually entered into a Microsoft Excel worksheet by the HMIS personnel.

The logbook has information on the medical registration number, date and time of patient's arrival, name, age, sex, region, admission date, diagnosis, mechanism of injury, injured body part, pathology, route of transfer from ward to ward, disposition, and disposition date.

## Study population and data collection

The study population included all new patients who presented to the ED department with evidence of injury as determined by clinicians. Patients with repeat visits to the ED due to the same injury or medical condition were excluded from the study. However, patients who had previously visited another healthcare facility or who had been referred to AaBET hospital were included in the study.

Data were collected through a retrospective record review of all patients seen in the ED department for 12 consecutive months from April 1, 2021 to March 30, 2022. The data collection tool was a standardized data collection form, based on recommendations by the World Health Organization (WHO) injury surveillance guideline [27]. Using the WHO's modified data collection form, data were retrieved and extracted from medical records in Microsoft Excel worksheets by HMIS personnel. After the data were extracted, patient names were deleted from the Microsoft Excel worksheet, and all de-identified data were kept confidential and not shared or disclosed to third parties. Then, the HMIS personnel delivered the data to the first author. Data retrieval and extraction were conducted using the existing resources at the hospital without additional manpower and resources.

Each patient was then assigned a unique study identifier. Patients'demographics (age, sex, and residence region), mechanism of injury, site of an injured body region, nature of injury, length of emergency stay, and disposition of patients were collected. Incomplete data were extracted by locating and retrieving the patients' medical records. When it was impossible to retrieve missing variables such as a missing diagnosis, the cases were excluded from the study. The medical records were accessed for research purposes from April 1, 2021 to March 30, 2022, and all data were fully anonymized. Therefore, the authors had no access to information that could identify individual participants during or after data collection.

## Definition and assessment criteria

An injury was any physical damage to the body caused by an intolerable level of energy. Injury cases were referred to a person who has been injured, regardless of the number of injuries sustained. The types of injury mechanisms based on the WHO classification included RTI, burns, falls, personal assault, machine injury, being stuck/hit by an object, animal bites, poisoning, alcohol toxicity, suicidal attempt, and unknown [27].

The number of all cases presented to the ED was recorded to determine the percentage of injury patients relative to the total number of patients visiting the hospital's ED during the study period. For patients with two or more injuries, the most severely injured body regions were considered in the report. The affected body parts were classified based on the affected

body region locations. These regions were head including the cranium region, eye, ear, mouth, nose, and face; trunk including back chest, abdomen and pelvis; upper limb; lower limb; and general regions (which included multiple locations on the skin, circulatory system, nervous system and the like). Injured patients with missing data on the mechanism of injury or other variables were categorized as 'unknown' and included in the study. However, those with missing diagnosis data were excluded.

In line with the WHO's classification of injury [27], which categorizes injuries as yellow/green (mild), orange (requires skilled care and treatment); and red (requires intensive medical/surgical care), the severity is believed to increase from the green to the red zone along the color spectrum. Neither the Injury Severity Score nor the Revised Trauma Scale was used to determine severity in the current study, and mortality from injury was defined as death in the ED zones.

The LoS in the ED refers to the total amount of time that injured patients spend in the emergency zones, ranging from minutes to hours (or even days) starting from when they log in [18, 28]. Patients who were observed for less than 24 hours or discharged from the ED on the same calendar day were considered to have one-day LoS [29]. According to the guidelines for the reform of hospital services in Ethiopia, patients' LoS in the ED should not exceed 24 hours. If necessary, transfer to inpatient should be made before the 24 hours for appropriate admission [30]. The patient's LoS was considered prolonged if it exceeded 24 hours. Disposition from the ED was categorized as discharged to home after treatment, transferred to another department within the hospital for admission, transferred to another hospital, absconded/left against medical advice, or died.

## Data quality control

Every effort was made to ensure data accuracy and veracity through data verification and validation by triangulating from HMIS and patient cards/charts. Data retrieval and extraction procedures were progressively conducted on weekdays, thus minimizing the risk of backlog and medical record loss.

## Data management and analysis

Data were entered into a Microsoft Excel worksheet. The data were exported to SPSS version 21.0 for analysis. Descriptive statistics were analyzed using frequency distribution, means, and standard deviations (SD). The primary outcome variable was LoS in the ED. Univariate and multivariable logistic regression were used to evaluate the association between LoS and independent variables. The independent variables were socio-demographic variables (sex, age, and residence), triaged zone, injury mechanisms, affected body parts, and nature of injuries. The significance between the categorical variables was assessed using the Pearson Chi-square test. Additionally, the collinearity test was performed by running collinearity diagnostics. The collinearity of each variable in this study was checked. Finally, variables with a p-value of less than 0.05 were considered statistically significant and were presented by the adjusted odds ratio (AOR) with a 95% confidence interval (CI) in the multivariable logistic regression analysis.

## Ethical approval

Ethical clearance for this study was obtained from the Institutional Review Board of the Ethiopian Public Health Association prior to the start of data collection (Ref. No: EPHA/06/039/21, Dated February 01, 2021). Permission to conduct this study was obtained from the appropriate chairs of the Departments/Units, including the hospital management board.

## Results

During one-year period, a total of 8055 patients with various diagnoses visited and were triaged at the ED. Of these cases, injuries accounted for 6991 (86.8%) cases. The majority of patients, 5694 (81.4%), were triaged to the yellow /green zone, followed by 758 (10.8%) to the orange zone, and, 539 (7.7%) to the red zone (Fig 1).

### Demographic characteristics and triage zones

The majority of injured patients in this study were male (74.1%, n = 5186). The mean age of the patients was 31.0 (SD = 17.2), ranging from 1 to 98 years. About half (50.4%, n = 3552) of the injured patients were from Addis Ababa city, followed by Oromia (40.2%, n = 2812). Of all age groups, the 18-29-year-olds (36.5%) had the highest number of recorded injury reports compared to other age groups. Similar patterns were observed in terms of the months of admission. Statistically significant differences in triage to emergency zones were found at a p-value <0.05. Sex, age, and residence region were also found to have statistically significant differences in patients' mode of triage to the yellow/green, orange, and red zones (Table 1).

### Mechanism of injury

The injury mechanism was captured for 5996 (85.7%) of injured patients (Table 2). The common mechanism of injury was RTI (n = 2693, 38.5%), followed by falls (n = 1523, 21.8%) and interpersonal violence (n = 1190, 17.0%). Burns (3.7%) and machinery accidents (3.0%) constituted 6.7% of all injuries. The injury mechanisms varied by sex and age group. RTIs were the leading mechanism of injury among age groups 18–29 (n = 1043, 40.8%), 30–44 (n = 729, 40.2%), and 45–59 (n = 309, 40.3%). Fall injuries were common among age groups 0–4 (n = 111, 37.6%), 5–17 (n = 322, 34.5%), and ≥60 years (n = 267, 43.1%). The majority of patients were from Addis Ababa and Oromia regions. About half of the patients in the red zone had RTI (Table 2).

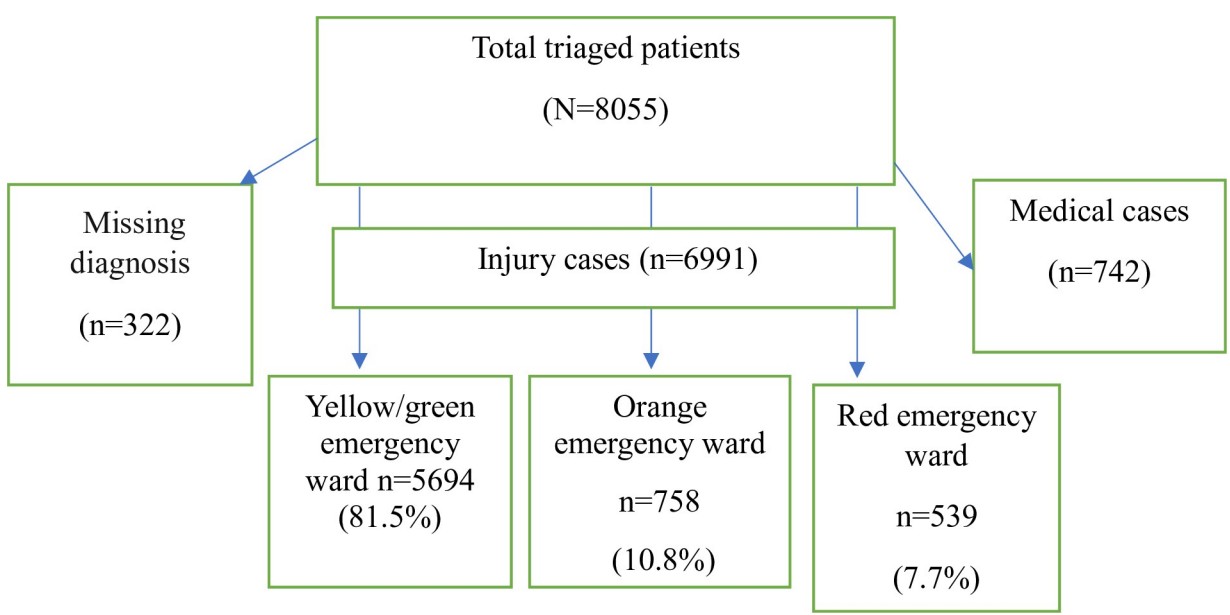

**Fig 1. The follow chart diagram of the study profile.**

**Table 1. Distribution of injured patients by emergency department zone.**

| Characteristics | Total, n (%) | Zone, n (%) | | | P-value |
|---|---|---|---|---|---|
| | | Yellow/green | Orange | Red | |
| **Sex** | | | | | 0.006 |
| Male | 5184 (74.2) | 4176 (73.3) | 575 (75.9) | 433 (80.3) | |
| Female | 1794 (25.7) | 1506 (26.4) | 182 (24.0) | 106 (19.7) | |
| Unknown | 13 (0.2) | 12 (0.20) | 1 (0.10) | 0 (0.0) | |
| **Age groups (year)** | | | | | |
| 1–4 | 295 (4.2) | 232 (4.1) | 41 (5.4) | 22 (4.1) | 0.001 |
| 5–17 | 934 (13.4) | 771 (13.5) | 96 (12.7) | 67 (12.4) | |
| 18–29 | 2555 (36.5) | 2115 (37.2) | 264 (34.8) | 176 (32.7) | |
| 30–44 | 1813 (25.9) | 1458 (25.6) | 196 (25.9) | 159 (29.5) | |
| 45–59 | 767 (11.0) | 623 (10.9) | 84 (11.1) | 60 (11.1) | |
| ≥60 | 619 (8.9) | 493 (8.7) | 75 (9.9) | 51 (9.5) | |
| Unknown | 8 (0.1) | 2 (0.0) | 2 (0.3) | 4 (0.7) | |
| **Regional residence** | | | | | <0.001 |
| Addis Ababa | 3522 (50.4) | 3105 (54.5) | 240 (31.7) | 177 (32.8) | |
| Oromia | 2812 (40.2) | 2123 (37.3) | 412 (54.4) | 277 (51.4) | |
| Amhara | 357 (5.1) | 254 (4.5) | 58 (7.7) | 45 (8.3) | |
| Other | 239 (3.4) | 179 (3.1) | 40 (5.3) | 20 (3.7) | |
| Unknown | 61 (0.9) | 33 (0.6) | 8 (1.1) | 20 (3.7) | |
| **Months of admission** | | | | | <0.001 |
| April-June, 2021 | 1774 (25.4) | 1435 (25.2) | 185 (24.4) | 154 (28.6) | |
| July–September, 2021 | 1605 (23.0) | 1283 (22.5) | 207 (27.3) | 115 (21.3) | |
| October -December,2021 | 1812 (25.9) | 1437 (25.2) | 239 (31.5) | 136 (25.2) | |
| January-March,2022 | 1789 (25.7) | 1539 (27.0) | 126 (16.6) | 133 (24.7) | |
| Unknown | 2 (0.01) | 0 | 1 (0.1) | 1 (0.2) | |
| **Total** | 6991(100%) | 5694 (81.5) | 758 (10.8) | 539 (7.7) | |

*Other includes Afar, Benishangul-Gumuz, Dire Dawa, Gambella, Harari, SNNPR, Somali regions, and military personnel

**Anatomic regions of injury.** Injured body regions were captured for 5873 (84%) of patients. The head body region was the most frequently recorded, with 2179 cases (31.2%) followed by the lower limb with 1365 cases (19.5%). Injuries to the upper limb accounted for 980 cases (14.0%), while injuries to the neck and truck accounted for 570 cases (8.2%), and injuries to general body regions accounted for 779 cases (11.1%) as found in the remaining records. However, in the cases of 1118 patients (16.0%), the location of their injury areas was unspecified or unknown (Table 3).

## Nature of injury

The injuries were classified based on the accident outcome into fractures; concussions and other internal injuries; contusions and crushing; dislocations, sprains and strains; and soft tissue injuries. Injury nature was captured for 6708 (96%) of patients. The most common nature of injuries was fracture, 2537 (36.3%), followed by concussion and other internal injuries, 1946 (27.8%) and soft tissue injuries, 1698 (24.3%) (Table 4).

**Table 2. Injury mechanism by patient characteristics.**

| Characteristics | Mechanisms of injury, n (%) | | | | | | | Total |
|---|---|---|---|---|---|---|---|---|
| | RTI | Fall | IPV | Burn | Machine | Others* | Unknown | |
| Sex | | | | | | | | |
| Male | 1958 (37.8) | 988 (19.1) | 1057 (20.4) | 136 (2.6) | 171 (3.3) | 75 (1.4) | 799 (15.4) | 5184 |
| Female | 734 (40.9) | 531 (29.6) | 129 (7.2) | 126 (7.0) | 36 (2.0) | 44 (2.5) | 194 (10.8) | 1794 |
| Unknown | 1 (7.7) | 4 (30.8) | 4 (30.8) | 0 | 0 | 2 (15.4) | 2 (15.4) | 13 |
| Age groups (year) | | | | | | | | |
| 0–4 | 52 (17.6) | 111 (37.6) | 3 (1.0) | 77 (26.1) | 4 (1.4) | 4 (1.4) | 44 (14.9) | 295 |
| 5–17 | 314 (33.6) | 322 (34.5) | 89 (9.5) | 49 (5.2) | 6 (0.6) | 21 (2.2) | 133 (14.2) | 934 |
| 18–29 | 1043 (40.8) | 293 (11.5) | 582 (22.8) | 83 (3.2) | 125 (4.9) | 53 (2.1) | 376 (14.7) | 2555 |
| 30–44 | 729 (40.2) | 321 (17.7) | 374 (20.0) | 37 (2.0) | 55 (3.0) | 30 (1.7) | 267 (14.7) | 1813 |
| 45–59 | 309 (40.3) | 208 (27.1) | 113 (14.7) | 10 (1.3) | 13 (1.7) | 7 (0.9) | 107 (14.0) | 767 |
| ≥60 | 242 (39.1) | 267 (43.1) | 27 (4.4) | 6 (1.0) | 4 (0.6) | 6 (1.0) | 67 (10.8) | 619 |
| Unknown | 4 (50.0) | 1 (12.5) | 2 (25.0) | 0 (0.0) | 0 (0.0) | 0 (0.0) | 1 (12.5) | 8 |
| Regional residence | | | | | | | | |
| Addis Ababa | 1250 (35.5) | 963 (27.3) | 495 (14.1) | 143 (4.1) | 118 (3.4) | 89 (2.5) | 464 (13.2) | 3522 |
| Oromia | 1210 (43.0) | 457 (16.3) | 522 (18.6) | 99 (3.5) | 79 (2.8) | 25 (0.9) | 420 (14.9) | 2812 |
| Amhara | 146 (40.9) | 60 (16.8) | 76 (21.3) | 11 (3.1) | 5 (1.4) | 3 (0.8) | 56 (15.7) | 357 |
| Other* | 78 (32.6) | 31 (13.0) | 79 (33.1) | 7 (2.9) | 2 (0.8) | 2 (0.8) | 40 (16.7) | 239 |
| Unknown | 9 (14.8) | 12 (19.7) | 18 (29.5) | 2 (3.3) | 3 (4.9) | 2 (3.3) | 15 (24.6) | 61 |
| Zone | | | | | | | | |
| Yellow/ green | 2082 (36.6) | 1448 (25.4) | 1068 (18.8) | 157 (2.8) | 204 (3.6) | 90 (1.6) | 645 (11.3) | 5694 |
| Orange | 337 (44.5) | 31 (4.1) | 30 (4.0) | 68 (9.0) | 1 (0.1) | 17 (2.2) | 274 (36.1) | 758 |
| Red | 274 (50.8) | 44 (8.2) | 92 (17.1) | 37 (6.9) | 2 (0.4) | 14 (2.6) | 76 (14.1) | 539 |
| **Total** | **2693 (38.5)** | **1523 (21.8)** | **1190 (17.0)** | **262 (3.7)** | **207(3.0)** | **121 (1.7)** | **995 (14.2)** | **6991 (100%)** |

*Other include animal bites, poisoning, alcohol toxicity, and suicidal attempt

RTI–Road Traffic Injury, IPV–Interpersonal Violence

## Length of stay in the emergency department

Of the 6991 patients, 5866 (83.9%) had recorded LoS (Table 5). About three-fourths of injured patients, 4432 (75.6%), stayed ≤ 24 hours at the ED. The magnitude of LoS greater than 24 hours in the ED was 24.4% (CI: 23.3–25.5). The mean (SD) length of stay for patients in the ED was 2.51 (±5.18) with a range of 1 to 181 days.

**Disposition from the emergency department.** Among the injured patients, the dispositions of 5669 (81.1%) patients were recorded. After receiving treatment in the ED zone, 3494 (50.0%) of the patients were discharged home, while 1712 (24.5%) were transferred to other departments within the hospital for further support. Additionally, 198 (2.8%) were referred to other hospitals, 164 (2.3%) patients left against medical advice from the ED, and 101 (1.8%, CI: 1.4–2.1) were reported as deceased.

Deaths included in this study were patients who were alive at the triage point and admitted for treatment in the emergency zones but died during treatment support. Among 101 deaths, the majority were male (n = 79, 78.2%), age groups of 18–29 (n = 35, 34.7%) and 30–44 (n = 33, 32.7%); and death occurred in the red zone, 78 (77.2%). More than half, (n = 59, 58.4%) of deaths were due to road traffic injuries specifically on the head and face body part (n = 57, 56.4%), and involved concussion and other internal injuries (n = 60, 59.4%) (Table 6).

**Table 3. Injured body regions by patient's characteristics.**

| Characteristics | Most common body regions injured, n (%) | | | | | | Total |
|---|---|---|---|---|---|---|---|
| | Head | Neck and Truck | upper limb | Lower Limb | General Injury | Unknown | |
| **Sex** | | | | | | | |
| Male | 1746 (33.7) | 417 (8.0) | 719 (13.9) | 971 (18.7) | 539 (10.4) | 792 (15.3) | 5184 |
| Female | 433 (24.1) | 150 (8.4) | 260 (14.5) | 394 (22.0) | 239 (13.3) | 318 (17.7) | 1794 |
| Unknown | 0 | 3 (23.1) | 1 (7.7) | 0 | 1 (7.7) | 8 (61.5) | 13 |
| **Age groups (year)** | | | | | | | |
| 0–4 | 77 (26.1) | 3 (1.0) | 34 (11.5) | 53 (18.0) | 90 (30.5) | 38 (12.9) | 295 |
| 5–17 | 291(31.2) | 43 (4.6) | 164 (17.6) | 164 (17.6) | 110 (11.8) | 162 (17.3) | 934 |
| 18–29 | 822 (32.2) | 231 (9.0) | 359 (14.1) | 390 (15.3) | 320 (12.5) | 433 (16.9) | 2555 |
| 30–44 | 582 (32.1) | 164 (9.0) | 239 (13.2) | 373 (20.6) | 174 (9.6) | 281 (15.5) | 1813 |
| 45–59 | 226 (29.5) | 72 (9.4) | 111 (14.5) | 192 (25.0) | 58 (7.6) | 108 (14.1) | 767 |
| ≥60 | 178 (28.8) | 56 (9.0) | 71 (11.5) | 192 (31.0) | 27 (4.4) | 95 (15.3) | 619 |
| Unknown | 3 (37.5) | 1 (12.5) | 2 (25.0) | 1 (12.5) | 0 (0.0) | 1 (12.5) | 8 |
| **Regional residences** | | | | | | | |
| Addis Ababa | 849 (24.1) | 216 (6.1) | 597 (17.0) | 723 (20.5) | 503 (14.3) | 634 (18.0) | 3522 |
| Oromia | 1108 (39.4) | 281 (10.0) | 317 (11.3) | 493 (17.5) | 217 (7.7) | 396 (14.1) | 2812 |
| Amhara | 125 (35.0) | 45 (12.6) | 29 (8.1) | 83 (23.2) | 29 (8.1) | 46 (12.9) | 357 |
| Other | 71 (29.7) | 19 (7.9) | 30 (12.6) | 62 (25.9) | 26 (10.9) | 31 (13.0) | 239 |
| Unknown | 26 (42.6) | 9 (14.8) | 7 (11.5) | 4 (6.6) | 4 (6.6) | 11 (18.0) | 61 |
| **Zones** | | | | | | | |
| Yellow/green | 1431 (25.1) | 399 (7.0) | 951 (16.1) | 1290 (22.7) | 634 (11.1) | 989 (17.4) | 5694 |
| Orange | 411 (54.2) | 114 (15.0) | 21 (2.8) | 57 (7.5) | 86 (11.3) | 69 (9.1) | 758 |
| Red | 337 (62.5) | 57 (10.6) | 8 (1.5) | 18 (3.3) | 59 (10.9) | 60 (11.1) | 539 |
| **Total** | **2179 (31.2)** | **570 (8.2)** | **980 (14.0)** | **1365 (19.5)** | **779 (11.1)** | **1118 (16.0)** | **6991 (100%)** |

## Factors associated with the emergency department length of stay

Univariate and multivariable logistic regression were used to assess the relationship between the independent variables and the prolonged LoS. The following factors were found to be significantly associated with the prolonged LoS: being male [AOR: 1.30, 95% CI (1.09–1.54)], age group ≥60 years [AOR: 1.38, 95% CI (1.03–1.85)]; residence outside Addis Ababa in Oromia [AOR: 1.61, 95% CI (1.38–1.88)], Amhara [AOR: 1.56, 95% CI (1.14–2.14)] and other [AOR: 3.93, 95% CI (2.73–5.64)]; injury mechanism of RTI [AOR: 2.43, 95% CI (1.19–4.94)], first triaged zones of orange [AOR: 3.03, 95% CI (2.40–3.83)] and red [AOR:3.37, 95% CI (2.65–4.28)]; fracture nature of injuries [AOR: 1.95, (1.34–2.83)]; and contusions and crushing [AOR: 2.63 (1.57–4.42)] (Table 7).

## Discussion

Our study revealed the characteristics of injuries, including mechanism, affected body parts, nature of injuries, LoS, and dispositions among patients who visited the ED. The majority of patients, who were injured were male, with the most common mechanism being road traffic accidents, resulting in head injuries and fractures nature of injury. About one-third of patients experienced a prolonged LoS. Factors that were significantly associated with prolonged LoS included being male, being in the age group ≥60 years; being residence outside Addis Ababa regions, experiencing road traffic accidents; being triaged in the orange and red zones, having fracture of injuries; and having contusions and crushing.

**Table 4. Nature of injury by patients'characteristics.**

| Characteristics | Nature of injuries, n (%) | | | | | | Total |
|---|---|---|---|---|---|---|---|
| | Fracture | Concussion and other internal injury | Contusions and crushing | Soft tissue injury | Sprain, strain, or dislocation | Unknown | |
| **Sex** | | | | | | | |
| Male | 1865 (36.0) | 1538 (29.7) | 154 (3.0) | 1198 (23.1) | 216 (4.2) | 213 (4.1) | 5184 |
| Female | 669 (37.3) | 406 (22.6) | 50 (2.8) | 495 (27.6) | 105 (5.9) | 69 (3.8) | 1794 |
| Unknown | 3 (23.1) | 2 (15.2) | 0 (0.0) | 5 (38.5) | 2 (15.4) | 1 (7.7) | 13 |
| **Age groups (year)** | | | | | | | |
| 0–4 | 77 (26.1) | 63 (21.4) | 2 (0.7) | 129 (43.6) | 12 (4.1) | 12 (4.1) | 295 |
| 5–17 | 347 (37.2) | 244 (26.1) | 17 (1.8) | 239 (25.6) | 52 (5.6) | 35 (3.7) | 934 |
| 18–29 | 809 (31.9) | 748 (29.3) | 87 (3.4) | 694 (27.2) | 121 (4.7) | 96 (3.8) | 2555 |
| 30–44 | 682 (37.6) | 526 (29.0) | 56 (3.1) | 408 (22.5) | 73 (4.0) | 68 (3.8) | 1813 |
| 45–59 | 316 (41.2) | 202 (26.3) | 29 (3.8) | 143 (18.6) | 41 (5.3) | 36 (4.7) | 767 |
| ≥60 | 304 (49.1) | 160 (25.8) | 12 (1.9) | 84 (13.6) | 24 (3.9) | 35 (5.7) | 619 |
| Unknown | 2 (25.0) | 3 (37.5) | 1 (12.5) | 1 (12.5) | 0 (0.0) | 1 (12.5) | 8 |
| **Regional residence** | | | | | | | |
| Addis Ababa | 1269 (36.0) | 771 (21.9) | 75 (2.1) | 211 (6.0) | 1071(30.4) | 125 (3.5) | 3522 |
| Oromia | 1005 (35.7) | 973 (34.6) | 109 (3.9) | 89 (3.2) | 513 (18.2) | 123 (4.4) | 2812 |
| Amhara | 149 (41.7) | 111 (31.1) | 14 (3.9) | 12 (3.4) | 52 (14.6) | 19 (5.3) | 357 |
| Other | 100 (41.8) | 64 (26.8) | 5 (2.1) | 10 (4.2) | 49 (20.5) | 11 (4.6) | 239 |
| Unknown | 14 (23.0) | 27 (44.3) | 1 (1.6) | 1 (1.6) | 13 (21.3) | 5 (8.2) | 61 |
| **Zones** | | | | | | | |
| Yellow/green | 2312 (40.6) | 1230 (21.6) | 120 (2.1) | 1528 (26.8) | 288 (5.1) | 216 (3.8) | 5694 |
| Orange | 151 (19.9) | 369 (48.7) | 63 (8.3) | 110 (14.5) | 21 (2.8) | 44 (5.8) | 758 |
| Red | 74 (13.7) | 347 64.48) | 21 (3.9) | 60 (11.1) | 14 (2.6) | 23 (4.3) | 539 |
| **Total** | **2537 (36.3)** | **1946 (27.8)** | **204 (2.9)** | **1698 (24.3)** | **323 (4.6)** | **283 (4.0)** | **6991 (100%)** |

The majority of patients who visited the ED with injuries were male and had a mean age of 31.0. However, most of the patients were male and fell into the age group of 18–29 years. Our findings are in line with studies conducted in Ethiopia [13, 17], Kenya [31, 32], Tanzania [33], Cameroon [34, 35] and Iran [36]. Evidence has shown that males tend to take greater risks and are less diligent in applying safety precautions compared to women [37]. These characteristics might make men more prone to injury incidents than women. Young adults might be engaged in multiple work-related and social activities with less attention to safety measures [38]. Additionally, these age groups might have more interaction with the environment in risk conditions, leading to riskier attitudes and significantly lower cognitive risk perceptions compared to older individuals [39].

These children and older age groups need support in guiding and care to prevent fall accidents. The current study found that fall injuries were common among those aged 0–4 and ≥60 years. This result is consistent with a community-based study conducted in Ethiopia [40]. The reason for this may be the way caregivers handle children [41] and the fact that older individuals might be prone to fall accidents in their daily activities [42].

The current study revealed that the common mechanism of injury was RTIs followed by falls. This finding was similarly reported by studies in Ethiopia [13], Kenya [32], Tanzania [33], Cameron [34, 35] and Iran [36]. The reason for this result might be due to unsafe transportation options, poor road infrastructure, personal behavior towards road safety, and inadequate policies to ensure road safety, which may contribute to the high burden of RTIs in

**Table 5. Length of stay in the emergency department by patients' characteristics.**

| Characteristics | Length of stay, n (%) | | | | | |
|---|---|---|---|---|---|---|
| | ≤ 24 hours | 2–3 days | 4–7 days | 8–14 days | 15–30 days | >30 days |
| **Sex** | | | | | | |
| Male | 3226 (72.8 | 461 (76.8) | 270 (76.9) | 259 (80.2) | 107 (79.3) | 18 (72.0) |
| Female | 1196 (27.0) | 138 (23.0) | 81 (23.1) | 64 (19.8) | 28 (20.7) | 7 (28.0) |
| Unknown | 10 (0.2) | 1 (0.2) | 0 (0.0) | 0 (0.0) | 0 (0.0) | 0 (0.0) |
| **Age groups (year)** | | | | | | |
| 0–4 | 200 (4.5) | 21 (3.5) | 11 (3.1) | 7 (2.2) | 2 (1.5) | 0 (0.0) |
| 5–17 | 599 (13.5) | 68 (11.3) | 47 (13.4) | 37 (11.5) | 9 (6.7) | 4 (16.0) |
| 18–29 | 1709 (38.6) | 228 (38.0) | 121 (34.5) | 113 (35.0) | 44 (32.6) | 6 (24.0) |
| 30–44 | 1137 (25.7) | 160 (26.7) | 88 (25.1) | 91 (28.2) | 43 (31.9) | 9 (36.0) |
| 45–59 | 450 (10.2) | 65 (10.8) | 50 (14.2) | 35 (10.8) | 15 (11.1) | 3 (12.0) |
| ≥60 | 332 (7.5) | 57 (9.5) | 34 (9.7) | 39 (12.1) | 22 (16.3) | 3 (12.0) |
| Unknown | 5 (0.1) | 1 (0.2) | 0 (0.0) | 1 (0.3) | 0 (0.0) | 0 (0.0) |
| **Regional residence** | | | | | | |
| Addis Ababa | 2433 (54.9) | 225 (37.5) | 127 (36.2) | 133 (41.2) | 50 (37.0) | 6 (24.0) |
| Oromia | 1664 (37.5) | 278 (46.3) | 179 (51.0) | 159 (49.2) | 68 (50.4) | 14 (56.0) |
| Amhara | 193 (4.4) | 35 (5.8) | 29 (8.3) | 15 (4.6) | 10 (7.4) | 3 (12.0) |
| Other | 106 (2.4) | 53 (8.8) | 14 (4.0) | 14 (4.3) | 6 (4.4) | 1 (4.0) |
| Unknown | 36 (0.8) | 9 (1.5) | 2 (0.6) | 2 (0.6) | 1 (0.7) | 1 (4.0) |
| **Zones** | | | | | | |
| Yellow/green | 3721 (84.0) | 353 (58.8) | 197 (56.1) | 238 (73.7) | 82 (60.7) | 12 (48.0) |
| Orange | 417 (9.4) | 137 (22.8) | 93 (26.5) | 44 (13.6) | 32 (23.7) | 3 (12.0) |
| Red | 294 (6.6) | 110 (18.3) | 61 (17.4) | 41 (12.7) | 21 (15.6) | 10 (40.0) |
| **Total** | **4432 (75.6)** | **600 (10.2)** | **351 (6.0)** | **323 (5.5)** | **135 (2.3)** | **25 (0.4)** |

developing countries [43]. However, this finding is inconsistent with a similar study conducted in Ethiopia [16], which reported intimate partner violence as the leading cause, followed by road traffic accidents. The disparity in the results might be due to differences in the study period. Severe injury cases were referred to the current study hospital, which is the only trauma center in Ethiopia during the study period. Another explanation could be the increasing trend of road traffic injuries in Ethiopia over the years leading to more reported cases during the study period compared to previous years in the country [44].

In line with a similar study in Cameroon [34], the majority of patients in the current study had head injuries followed by neck and trunk as a result of accidents. The reason for this might be the mechanism of injury with road traffic and falls being more commonly reported causes of head injuries [45, 46]. Furthermore, fractures were the most common type of injury observed in the current study. This finding is consistent with similar studies in Ethiopia [16, 17] and Tanzania [29]. The explanation might be that RTIs are associated with more severe outcomes, such as fractures due to the significant force exerted on the body during accidents [46].

The results of this study showed that the majority of injured patients survived during the study period and were discharged home. However, 101 (1.8%, CI: 1.4–2.1) patients' disposition was death. This is consistent with an Iranian study that reported a death rate of 1.5% [36]. However, our finding is lower than the study in Tanzania, which reported a death rate of 5.4% [29] and higher than the findings in Cameron (0.4% and 1.02%) [34, 35]. The disparity in findings among the studies might be due to differences in facilities for care or service [47], leading mechanisms of injuries, body parts affected, and severity of injuries [46].

**Table 6. Disposition from the emergency department by patient characteristics.**

| Characteristics | Disposition, n (%) | | | | | | Total |
|---|---|---|---|---|---|---|---|
| | Discharged home | Admitted | Referred | Absconded | Died | Unknown | |
| **Sex** | | | | | | | |
| Male | 2500 (48.2) | 1314 (25.3) | 161 (3.1) | 132 (2.5) | 79 (1.5) | 998 (19.3) | 5186 |
| Female | 987 (55.0) | 396 (22.1) | 37 (2.1) | 32 (1.8) | 22 (1.2) | 320 (17.8) | 1796 |
| Unknown | 7 (53.8) | 2 (15.4) | 0 (0.0) | 0 (0.0) | 0 (0.0) | 4 (30.8) | 13 |
| **Age groups (year)** | | | | | | | |
| 0–4 | 129 (43.7) | 93 (31.5) | 8 (2.7) | 3 (1.0) | 2 (0.7) | 60 (20.3) | 295 |
| 5–17 | 456 (48.8) | 232 (24.8) | 19 (2.0) | 16 (1.7) | 9 (1.0) | 202 (21.6) | 934 |
| 18–29 | 1394 (54.6) | 591 (23.1) | 72 (2.8) | 59 (2.3) | 35 (1.4) | 404 (15.8) | 2555 |
| 30–44 | 901 (49.7) | 447 (24.7) | 48 (2.6) | 48 (2.6) | 33 (1.8) | 336 (18.5) | 1813) |
| 45–59 | 368 (48.0) | 176 (22.9) | 32 (4.2) | 13 (1.7) | 9 (1.2) | 169 (22.0) | 767 |
| ≥60 | 244 (39.4) | 169 (27.3) | 19 (3.1) | 25 (4.0) | 13 (2.1) | 149 (24.1) | 619 |
| Unknown | 2 (25.0) | 4 (50.0) | 0 (0.0) | 0 (0.0) | 0 (0.0) | 2 (25.0) | 8 |
| **Regional residence** | | | | | | | |
| Addis Ababa | 1968 (55.9) | 737 (20.9) | 71 (2.0) | 79 (2.2) | 33 (0.9) | 634 (18.0) | 3522 |
| Oromia | 1297 (46.1) | 771 (27.4) | 81 (2.9) | 64 (2.3) | 53 (1.9) | 546 (19.4) | 2812 |
| Amhara | 138 (38.7) | 111 (31.1) | 13 (3.6) | 10 (2.8) | 10 (2.8) | 75 (21.0) | 357 |
| Other | 77 (32.2) | 67 (28.0) | 30 (12.6) | 9 (3.8) | 3 (1.3) | 53 (22.2) | 239 |
| Unknown | 14 (23.0) | 26 (42.6) | 3 (4.9) | 2 (3.3) | 2 (3.3) | 14 (23.0) | 61 |
| **Zones** | | | | | | | |
| Yellow/green | 3212 (56.4) | 1009 (17.7) | 130 (2.3) | 145 (2.5) | 4 (0.1) | 1194 (21.0) | 5694 |
| Orange | 259 (34.2) | 317 (41.8) | 52 (6.9) | 15 (2.0) | 19 (2.5) | 96 (12.7) | 758 |
| Red | 23 (4.3) | 386 (71.6) | 16 (3.0) | 4 (0.7) | 78 (14.5) | 32 (5.9) | 539 |
| **Total** | **3494 (50.0)** | **1712 (24.5)** | **198 (2.8)** | **164 (2.3)** | **101 (1.4)** | **1322 (18.9)** | **6991 (100%)** |

Prolonged stays in the ED may have a negative impact on patient outcomes and it is a poor service performance indicator [48, 49]. According to the Ethiopian Hospital Services Transformation Guideline [36], patients should not stay in the ED for more than 24 hours. In line with another study [50], the current mean (SD) LoS was 2.51 (±5.18) and greater than the national guideline of less than one day. The magnitude of prolonged LoS greater than 24 hours in the ED was 24.4% (CI: 23.3–25.5). This finding is lower than another study in South Ethiopia (91.5%) [28] and North Ethiopia, 46.5% [18].The difference in LoS between the current finding and other studies might be due to differences in healthcare settings. The current study hospital is known for services only for trauma care, while the other is a comprehensive and specialized hospitals.

Males had significantly higher odds of prolonged LoS as compared to females. This finding is also reported in another study, where gender has been found to influence the likelihood of male trauma victims staying in ED [51]. The reason for this might be due to the limitation of resources in hospitals, where priorities might be given to females for services resulting in different LoS. Furthermore, evidence shows that LoS increases significantly with age [48]. This is in line with our study, which found that the age group of ≥60 years has a greater odd staying more than 24 hours compared to the age of 0–17 years. The reason for this might be that older age groups require more diagnostic and therapeutic options, which might not be available for immediate care in the hospital [18, 28, 50].

Patient residence outside of Addis Ababa and triaged to the red zone were significantly identified factors that were more likely to result in prolonged LoS. The reason might be that

**Table 7. Factors associated with prolonged LoS at the emergency department.**

| Characteristics | Length of stay | | Crude OR (95% CI) | Adjusted OR (95% CI) |
|---|---|---|---|---|
| | ≤24 hours | >24 hours | | |
| **Sex** (n = 5855) | | | | |
| Male | 3226 (74.3) | 1115 (25.7) | 1.30 (1.13–1.50) * | 1.30 (1.09–1.54) ** |
| Female | 1196 (79.0) | 318 (21.0) | 1 | 1 |
| **Age group in year** *(n = 5859* | | | | |
| 0–17 | 799 (79.5) | 206 (20.5) | 1 | 1 |
| 18–29 | 1709 (76.9) | 512 (23.1) | 1.16 (0.97–1.39) | 1.01 (0.81–1.27) |
| 30–44 | 1137 (74.4) | 391 (25.6) | 1.33 (1.10–1.62) | 1.10 (0.87–1.39) |
| 45–59 | 450 (72.8) | 168 (27.2) | 1.45 (1.15–1.83) | 1.21 (0.92–1.61) |
| ≥60 | 332 (68.2) | 155 (31.8) | 1.81 (1.42–2.31) | 1.38 (1.03–1.85) ** |
| **Regional residence** *(n = 5815)* | | | | |
| Addis Ababa | 2433 (81.8) | 541 (18.2) | 1 | 1 |
| Oromia | 1664 (70.4) | 698 (29.6) | 1.89 (1.66–2.15) | 1.61 (1.38–1.88) ** |
| Amhara | 193 (67.7) | 92 (32.3) | 2.14 (1.64–2.80) | 1.56 (1.14–2.14) ** |
| Other | 106 (54.7) | 88 (45.3) | 3.73 (2.77–5.03) | 3.93 (2.73–5.64) ** |
| **Mechanism of injury** *(n = 5066)* | | | | |
| RTI | 1886 (70.6) | 787 (29.4) | 2.86 (1.59–5.14) | 2.43 (1.19–4.94) ** |
| Fall | 808 (79.5) | 208 (20.5) | 1.76 (0.97–3.22) | 1.72 (0.84–3.51) |
| IPV | 717 (79.5) | 185 (20.5) | 1.77 (0.97–3.23) | 1.62 (0.79–3.33) |
| Burn | 167 (76.6) | 51 (23.4) | 2.01 (1.08–4.05) | 2.08 (0.97–4.46) |
| Machine | 136 (87.7) | 19 (12.3) | 0.91 (0.45–2.03) | 1.06 (0.44–2.57) |
| Animal bites/ poisoning/alcohol toxicity/ suicidal attempt | 89 (87.3) | 13 (12.7) | 1 | 1 |
| **Triage Zone** *(n = 5866)* | | | | |
| Yellow/green | 3721 (80.8) | 882 (19.2) | 1 | 1 |
| Orange | 417 (57.4) | 309 (42.6) | 3.13 (2.65–3.69) * | 3.03 (2.40–3.83) ** |
| Red | 294 (54.7) | 243 (45.3) | 3.49 (2.90–4.20) * | 3.37 (2.65–4.28) ** |
| **Severely injured body part** *(n = 5044)* | | | | |
| Head | 1404 (71.7) | 553 (28.3) | 2.17 (1.73–2.72) * | 0.89 (0.55–1.43) |
| Neck and truck | 297 (60.1) | 197 (39.9) | 3.65 (2.79–4.79) * | 1.27 (0.77–2.09) |
| Upper limb | 661 (80.3) | 162 (19.7) | 1.35 (1.04–1.76) * | 0.94 (0.59–1.51) |
| Lower limb | 725 (69.6) | 316 (30.4) | 2.40 (1.89–3.06) | 1.20 (0.75–1.92) |
| General injury | 617 (84.6) | 112 (15.4) | 1 | 1 |
| **Nature of injury (n = 5723)** | | | | |
| Fracture | 1420 (70.5) | 595 (29.5) | 3.10 (2.59–3.71) * | 1.95 (1.34–2.83) ** |
| Concussions and other internal injuries | 1208 (71.1) | 492 (28.9) | 3.01 (2.50–3.63) * | 1.42 (0.95–2.14) |
| Contusions and crushing | 103 (54.2) | 87 (45.8) | 6.24 (4.52–8.63) * | 2.63 (1.57–4.42) ** |
| Acute poisonings, Sprain, strain or dislocation | 214 (83.3) | 43 (16.7) | 1.49 (1.04–2.13) * | 1.24 (0.76–2.03) |
| Soft Tissue injuries | 1375 (88.1) | 186 (11.9) | 1 | 1 |

* p-value <0.05 at OR = Odds Ratio, CI = Confidence Interval

** p-value <0.05, at AOR = Adjusted Odds Ratio

patients from outside Addis Ababa might stay at ED until transportation arrangements are made to reduce the accommodation costs, including hotel expenses. Furthermore, patients with more severe cases are triaged to the red zone while those with less severe injuries are placed in the Yellow/Green zones. As a result, patients in the red emergency zone might have longer stays compared to patients in the yellow/green zone.

It is important to consider the mechanism of injury to understand the nature of injuries and the required diagnosis, including radiology services [52]. The current study revealed that prolonged LoS was more likely among patients with RTI compared to patients with animal bites, poisoning, alcohol toxicity, and suicidal attempts. The justification for this report is that road traffic accidents are the most common traumatic mechanism of injury that is associated with severe forms of injury [52]. In addition, fractures; and injuries resulting in contusions and crushing were significantly associated with prolonged LoS compared to soft tissue injuries. This finding is consistent with studies conducted in Iran [51]. These associations might be explained by the serious nature of these injuries like fractures; contusions and crushing injuries may require more care and support due to their more serious nature [53].

## Limitations

First, our study was based on secondary data and cannot provide sufficient evidence of causality concerning injury severity and LoS in EDs with their associated factors. Second, we analyzed the data which was some incomplete and from emergency patients in Trauma hospital. Therefore; these findings might not represent other types of hospitals. However, this study's findings are based on a large data set and ultimately provide helpful information on the characteristics of injuries and LoS in EDs in similar countries with limited data conditions.

## Conclusions

The trauma registration has provided an opportunity to understand the country's burden of injury. Injuries are one of the major causes of morbidity among patients in the trauma hospital in Ethiopia. The leading mechanism of injury was road traffic accidents, which, predominantly affected males and people aged 15–44 years age groups. Head injuries and fractures were the most commonly reported outcome. The mean LoS was longer compared to the Ethiopian Hospital Services Guideline. Gender, age group ($\geq$60) years, residence far from the hospital, injury mechanism, first triaged zones, and injury nature were significantly identified as factors to prolonged LoS in ED.

Intervention measures with an emphasis on road traffic accidents, falls and IPV are needed to reduce the hospital burden with patients; and recommendation on strengthen the ED of the hospital to minimize the LoS. The prolonged LoS alarms the responsible government organizations and other stakeholders to maintain the necessary infrastructure in Ethiopia. Furthermore, research in a broader context based with primary data is recommended.

## Acknowledgments

The authors would like to thank the clinical and HMIS staff at AaBET hospital for providing the necessary technical, and logistical support for this study. We would also like to thank everyone who works in the medical record-keeping room.

## Author Contributions

**Conceptualization:** Hailemichael Mulugeta, Ayalew Zewdie, Wakgari Deressa.

**Data curation:** Hailemichael Mulugeta, Wakgari Deressa.

**Formal analysis:** Hailemichael Mulugeta.

**Investigation:** Hailemichael Mulugeta.

**Methodology:** Hailemichael Mulugeta, Wakgari Deressa.

**Software:** Hailemichael Mulugeta.

**Supervision:** Hailemichael Mulugeta, Ayalew Zewdie.

**Visualization:** Wakgari Deressa.

**Writing – original draft:** Hailemichael Mulugeta, Wakgari Deressa.

**Writing – review & editing:** Ayalew Zewdie, Tesfaye Getachew, Wakgari Deressa.

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
