## [Decision Letter · Decision Letter 0]

11 Jun 2024

PONE-D-23-40594Epidemiology of injury and patients` length of stay at emergency department of a trauma hospital in Addis Ababa, EthiopiaPLOS ONE

Dear Dr. Mulugeta,

Thank you for submitting your manuscript to PLOS ONE. After careful consideration, we feel that it has merit but does not fully meet PLOS ONE’s publication criteria as it currently stands. Therefore, we invite you to submit a revised version of the manuscript that addresses the points raised during the review process. Please reveiew the reviewers comments carefully. Please review and proofread the manuscript before submitting a revised version.

We look forward to receiving your revised manuscript.

Kind regards,

Alaa Oteir, PhD

Academic Editor

PLOS ONE

Journal Requirements:

3. Thank you for stating the following in your Competing Interests section: "None Declared"

Reviewers' comments:

Reviewer's Responses to Questions

**Comments to the Author**

1. Is the manuscript technically sound, and do the data support the conclusions?

Reviewer #1: Partly

2. Has the statistical analysis been performed appropriately and rigorously? 

Reviewer #1: Yes

3. Have the authors made all data underlying the findings in their manuscript fully available?

Reviewer #1: Yes

4. Is the manuscript presented in an intelligible fashion and written in standard English?

Reviewer #1: No

5. Review Comments to the Author

Reviewer #1: • The manuscript has a lot of editorial issues (incomplete ideas, repeated words, grammar, punctuation etc…)

• Please be consistent when you present your findings. For example, using both the frequency and (percentage) or only the percentage

• The authors mention that “neither the 210 Injury Severity Score nor the Revised Trauma Scale was used to determine severity”. if that is the case, what specific criteria do they use to label yellow/green, orange, and red categories?

• I am not clear on the objective of this manuscript. The outcome of this study was the length of stay at ED. The independence variables were socio-demographic variables (sex, age, and residence), triaged zone, mechanisms of injury, affected body parts, and nature of injuries. All the mentioned independent variables are linked with the patient, not the facility and the service. But in most of the cases, the length of stay in the ED depends on the capacity of the facilities, the service, and infrastructure-related problems and this needs additional investigation using a qualitative approach.

• The authors mention that the mean (SD) length of patients who stayed in the emergency department was 2.51 (5.18) with a range of 1 to 181 days. Why do they stay at the ED for a long time? They should be transferred either to a different ward or to other facilities for better care. According to Ethiopian hospital guidelines, patients requiring emergency services should be kept at the ED for a maximum of 24 h.

• I think the study should be modified to the length of stay at the facility, not at the ED

6. PLOS authors have the option to publish the peer review history of their article (what does this mean?). If published, this will include your full peer review and any attached files.

Reviewer #1: No

---

## [Author Response · Author response to Decision Letter 0]

21 Jun 2024

We would like to thank the reviewers for the time and effort put on to revise our manuscript in detail. We believe that the comments have identified important areas which required improvement. After completion of the suggested edits, the revised manuscript has benefitted from an improvement in the overall presentation and clarity.The following are the response to your issues.

Issue 1: The manuscript has a lot of editorial issues (incomplete ideas, repeated words, grammar, punctuation etc…) 

Response: Accepted the comment: Thank you for the critical observation. We made a thorough edition throughout the document.

Issue 2: Please be consistent when you present your findings for example using both the frequency and (percentage) or only the percentage 

Response: Accepted the comment

Issue 3: The authors mention that “neither the 210 Injury Severity Score nor the Revised Trauma Scale was used to determine severity”. if that is the case, what specific criteria do they use to label yellow/green, orange, and red categories?

Response: Thank you for the critical observation. We agreed in your issues

The Trauma Hospital used color coding for departmentalizing the emergency department (ED) room. The ED has four zones (or wards) to treat patients according to the Emergency Severity Index scale (1). These zones are identified by color-coded triage zones, which include green, yellow, orange, and red. In line with the WHO's classification of injury (2), which categorizes injuries as yellow/green (mild), orange (requires skilled care and treatment), and red (requires intensive medical/surgical care). The severity is believed to increase from the green to the red zone along the color spectrum. We used secondary data with incomplete data in the form of Injury Severity Score or the Revised Trauma Scale. However, we categorized based on severity according to the Emergency Severity Index scale of color coding.

Issue 4: I am not clear on the objective of this manuscript the outcome of this study was the length of stay at ED. The independence variables were sociodemographic variables (sex, age, and residence), triaged zone, mechanisms of injury, affected body parts, and nature of injuries. All the mentioned independent variables are linked with the patient, not the facility and the service. But in most of the cases, the length of stay in the ED depends on the capacity of the facilities, the service, and infrastructure-related problems and this needs additional investigation using a qualitative approach. 

Response: Thank you for the critical observation. We agreed in your issues.

This study aimed to describe injuries and identify factors associated with patients' prolonged lengths of stay (LoS) in the Emergency Department (ED). The findings of this study are based on a large dataset and provide information about the magnitude of injury burden, characteristics of injury in individuals, mechanism, and outcome. LoS was analyzed based on existing data due to record availability. However, we acknowledge the limitations of our study and recommend further research in a broader context based on primary data.

Issue 5: The authors mention that the mean (SD) length of patients who stayed in the emergency department was 2.51 (5.18) with a range of 1 to 181 days. Why do they stay at the ED for a long time? They should be transferred either to a different ward or to other facilities for better care. According to Ethiopian hospital guidelines, patients requiring emergency services should be kept at the ED for a maximum of 24 h.

Response: Thank you for the critical observation. We reviewed your concerns and have used the existing records. However, we also cross-checked the data with physicians and other workers in the ED. The reasons why patients stay for a long period are as follows:

1. Patient load in other departments, which leads to unavailability of beds.

2. Shortage of diagnostic tools.

3. There were patients without a responsible proxy or family members. These patients remained in the ED even after completing their treatment until a nonprofit /charity organization agreed to provide shelter and food support outside of the hospital.

Issue 6: I think the study should be modified to the length of stay at the facility, not at the ED

Response: Thank you for the critical observation. 

However, we emphasized the burden of patients in the ED, not in all departments in the hospitals. This is because the ED is the first area where lives are saved through appropriate support and linking to the right services at the right department.

Issue 1: The manuscript has a lot of editorial issues (incomplete ideas, repeated words, grammar, punctuation etc…) 

Response: Accepted the comment: Thank you for the critical observation. We made a thorough edition throughout the document.

Issue 2: Please be consistent when you present your findings for example using both the frequency and (percentage) or only the percentage 

Response: Accepted the comment

Issue 3: The authors mention that “neither the 210 Injury Severity Score nor the Revised Trauma Scale was used to determine severity”. if that is the case, what specific criteria do they use to label yellow/green, orange, and red categories?

Response: Thank you for the critical observation. We agreed in your issues

The Trauma Hospital used color coding for departmentalizing the emergency department (ED) room. The ED has four zones (or wards) to treat patients according to the Emergency Severity Index scale (1). These zones are identified by color-coded triage zones, which include green, yellow, orange, and red. In line with the WHO's classification of injury (2), which categorizes injuries as yellow/green (mild), orange (requires skilled care and treatment), and red (requires intensive medical/surgical care). The severity is believed to increase from the green to the red zone along the color spectrum. We used secondary data with incomplete data in the form of Injury Severity Score or the Revised Trauma Scale. However, we categorized based on severity according to the Emergency Severity Index scale of color coding.

Issue 4: I am not clear on the objective of this manuscript the outcome of this study was the length of stay at ED. The independence variables were sociodemographic variables (sex, age, and residence), triaged zone, mechanisms of injury, affected body parts, and nature of injuries. All the mentioned independent variables are linked with the patient, not the facility and the service. But in most of the cases, the length of stay in the ED depends on the capacity of the facilities, the service, and infrastructure-related problems and this needs additional investigation using a qualitative approach. 

Response: Thank you for the critical observation. We agreed in your issues.

This study aimed to describe injuries and identify factors associated with patients' prolonged lengths of stay (LoS) in the Emergency Department (ED). The findings of this study are based on a large dataset and provide information about the magnitude of injury burden, characteristics of injury in individuals, mechanism, and outcome. LoS was analyzed based on existing data due to record availability. However, we acknowledge the limitations of our study and recommend further research in a broader context based on primary data.

Issue 5: The authors mention that the mean (SD) length of patients who stayed in the emergency department was 2.51 (5.18) with a range of 1 to 181 days. Why do they stay at the ED for a long time? They should be transferred either to a different ward or to other facilities for better care. According to Ethiopian hospital guidelines, patients requiring emergency services should be kept at the ED for a maximum of 24 h.

Response: Thank you for the critical observation. We reviewed your concerns and have used the existing records. However, we also cross-checked the data with physicians and other workers in the ED. The reasons why patients stay for a long period are as follows:

1. Patient load in other departments, which leads to unavailability of beds.

2. Shortage of diagnostic tools.

3. There were patients without a responsible proxy or family members. These patients remained in the ED even after completing their treatment until a nonprofit /charity organization agreed to provide shelter and food support outside of the hospital.

Issue 6: I think the study should be modified to the length of stay at the facility, not at the ED

Response: Thank you for the critical observation. 

However, we emphasized the burden of patients in the ED, not in all departments in the hospitals. This is because the ED is the first area where lives are saved through appropriate support and linking to the right services at the right department.

Issue 1: The manuscript has a lot of editorial issues (incomplete ideas, repeated words, grammar, punctuation etc…) 

Response: Accepted the comment: Thank you for the critical observation. We made a thorough edition throughout the document.

Issue 2: Please be consistent when you present your findings for example using both the frequency and (percentage) or only the percentage 

Response: Accepted the comment

Issue 3: The authors mention that “neither the 210 Injury Severity Score nor the Revised Trauma Scale was used to determine severity”. if that is the case, what specific criteria do they use to label yellow/green, orange, and red categories?

Response: Thank you for the critical observation. We agreed in your issues

The Trauma Hospital used color coding for departmentalizing the emergency department (ED) room. The ED has four zones (or wards) to treat patients according to the Emergency Severity Index scale (1). These zones are identified by color-coded triage zones, which include green, yellow, orange, and red. In line with the WHO's classification of injury (2), which categorizes injuries as yellow/green (mild), orange (requires skilled care and treatment), and red (requires intensive medical/surgical care). The severity is believed to increase from the green to the red zone along the color spectrum. We used secondary data with incomplete data in the form of Injury Severity Score or the Revised Trauma Scale. However, we categorized based on severity according to the Emergency Severity Index scale of color coding.

Issue 4: I am not clear on the objective of this manuscript the outcome of this study was the length of stay at ED. The independence variables were sociodemographic variables (sex, age, and residence), triaged zone, mechanisms of injury, affected body parts, and nature of injuries. All the mentioned independent variables are linked with the patient, not the facility and the service. But in most of the cases, the length of stay in the ED depends on the capacity of the facilities, the service, and infrastructure-related problems and this needs additional investigation using a qualitative approach. 

Response: Thank you for the critical observation. We agreed in your issues.

This study aimed to describe injuries and identify factors associated with patients' prolonged lengths of stay (LoS) in the Emergency Department (ED). The findings of this study are based on a large dataset and provide information about the magnitude of injury burden, characteristics of injury in individuals, mechanism, and outcome. LoS was analyzed based on existing data due to record availability. However, we acknowledge the limitations of our study and recommend further research in a broader context based on primary data.

Issue 5: The authors mention that the mean (SD) length of patients who stayed in the emergency department was 2.51 (5.18) with a range of 1 to 181 days. Why do they stay at the ED for a long time? They should be transferred either to a different ward or to other facilities for better care. According to Ethiopian hospital guidelines, patients requiring emergency services should be kept at the ED for a maximum of 24 h.

Response: Thank you for the critical observation. We reviewed your concerns and have used the existing records. However, we also cross-checked the data with physicians and other workers in the ED. The reasons why patients stay for a long period are as follows:

1. Patient load in other departments, which leads to unavailability of beds.

2. Shortage of diagnostic tools.

3. There were patients without a responsible proxy or family members. These patients remained in the ED even after completing their treatment until a nonprofit /charity organization agreed to provide shelter and food support outside of the hospital.

Issue 6: I think the study should be modified to the length of stay at the facility, not at the ED

Response: Thank you for the critical observation. 

However, we emphasized the burden of patients in the ED, not in all departments in the hospitals. This is because the ED is the first area where lives are saved through appropriate support and linking to the right services at the right department.

Issue 1: The manuscript has a lot of editorial issues (incomplete ideas, repeated words, grammar, punctuation etc…) 

Response: Accepted the comment: Thank you for the critical observation. We made a thorough edition throughout the document.

Issue 2: Please be consistent when you present your findings for example using both the frequency and (percentage) or only the percentage 

Response: Accepted the comment

Issue 3: The authors mention that “neither the 210 Injury Severity Score nor the Revised Trauma Scale was used to determine severity”. if that is the case, what specific criteria do they use to label yellow/green, orange, and red categories?

Response: Thank you for the critical observation. We agreed in your issues

The Trauma Hospital used color coding for departmentalizing the emergency department (ED) room. The ED has four zones (or wards) to treat patients according to the Emergency Severity Index scale (1). These zones are identified by color-coded triage zones, which include green, yellow, orange, and red. In line with the WHO's classification of injury (2), which categorizes injuries as yellow/green (mild), orange (requires skilled care and treatment), and red (requires intensive medical/surgical care). The severity is believed to increase from the green to the red zone along the color spectrum. We used secondary data with incomplete data in the form of Injury Severity Score or the Revised Trauma Scale. However, we categorized based on severity according to the Emergency Severity Index scale of color coding.

Issue 4: I am not clear on the objective of this manuscript the outcome of this study was the length of stay at ED. The independence variables were sociodemographic variables (sex, age, and residence), triaged zone, mechanisms of injury, affected body parts, and nature of injuries. All the mentioned independent variables are linked with the patient, not the facility and the service. But in most of the cases, the length of stay in the ED depends on the capacity of the facilities, the service, and infrastructure-related problems and this needs additional investigation using a qualitative approach. 

Response: Thank you for the critical observation. We agreed in your issues.

This study aimed to describe injuries and identify factors associated with patients' prolonged lengths of stay (LoS) in the Emergency Department (ED). The findings of this study are based on a large dataset and provide information about the magnitude of injury burden, characteristics of injury in individuals, mechanism, and outcome. LoS was analyzed based on existing data due to record availability. However, we acknowledge the limitations of our study and recommend further research in a broader context based on primary data.

Issue 5: The authors mention that the mean (SD) length of patients who stayed in the emergency department was 2.51 (5.18) with a range of 1 to 181 days. Why do they stay at the ED for a long time? They should be transferred either to a different ward or to other facilities for better care. According to Ethiopian hospital guidelines, patients requiring emergency services should be kept at the ED for a maximum of 24 h.

Response: Thank you for the critical observation. We reviewed your concerns and have used the existing records. However, we also c

---

## [Decision Letter · Decision Letter 1]

22 Aug 2024

Injury Epidemiology and Emergency Department Length of Stay in Trauma Hospital in Addis Ababa, Ethiopia

PONE-D-23-40594R1

Dear Dr. Hailemichael Mulugeta,

We’re pleased to inform you that your manuscript has been judged scientifically suitable for publication and will be formally accepted for publication once it meets all outstanding technical requirements.

Kind regards,

Alaa Oteir, PhD

Academic Editor

PLOS ONE

Additional Editor Comments (optional):

Reviewers' comments:

Reviewer's Responses to Questions

**Comments to the Author**

1. If the authors have adequately addressed your comments raised in a previous round of review and you feel that this manuscript is now acceptable for publication, you may indicate that here to bypass the “Comments to the Author” section, enter your conflict of interest statement in the “Confidential to Editor” section, and submit your "Accept" recommendation.

Reviewer #1: All comments have been addressed

2. Is the manuscript technically sound, and do the data support the conclusions?

Reviewer #1: Yes

3. Has the statistical analysis been performed appropriately and rigorously? 

Reviewer #1: Yes

4. Have the authors made all data underlying the findings in their manuscript fully available?

Reviewer #1: Yes

5. Is the manuscript presented in an intelligible fashion and written in standard English?

Reviewer #1: Yes

6. Review Comments to the Author

Reviewer #1: The authors address all my comments. Hence I accepted their reviewed manuscript and now you can proceed the next step.

7. PLOS authors have the option to publish the peer review history of their article (what does this mean?). If published, this will include your full peer review and any attached files.

Reviewer #1: **Yes: **Teferi Abegaz

---

## [Editor Report · Acceptance letter]

6 Sep 2024

PONE-D-23-40594R1 

PLOS ONE

Dear Dr. Mulugeta, 

I'm pleased to inform you that your manuscript has been deemed suitable for publication in PLOS ONE. Congratulations! Your manuscript is now being handed over to our production team.

Kind regards, 

on behalf of

Dr. Alaa Oteir 

Academic Editor

PLOS ONE